# Pseudo-GT Driven Region-Constrained Black-Box Attack on Semantic Models for Autonomous Driving

## Abstract

Semantic segmentation models have been widely adopted in various domains, including safety-critical applications such as autonomous driving, where they play a pivotal role in enabling accurate scene understanding and decision-making. However, despite their utility and prevalence, these models remain vulnerable to targeted adversarial perturbations, which can compromise their reliability in real-world deployments. To highlight this challenge, we propose a two-stage attack framework that crafts structured perturbations confined to the central vehicle region, inducing misclassifications while preserving background semantics. First, we generate a pseudo-ground-truth segmentation by inpainting the detected vehicle mask within the central third of the image, enabling the attacker to anticipate the model's response if the target class were absent. Second, we optimize an $\ell_\infty$-bounded perturbation via a hybrid loss combining mean-squared error to the pseudo-ground-truth, total variation regularization for spatial coherence, and a class-wise IoU loss to degrade segmentation across all non-target classes. Finally, we refine the attack using region-specific cross-entropy losses to simultaneously mislead vehicle pixels toward surrounding classes and maintain background consistency. Evaluated on the Cityscapes dataset, our attack achieves over 92% predicted mask accuracy for the target zone and over 93% background preservation elsewhere with Segformer model. These results demonstrate that spatial constraints cannot prevent powerful region-based attacks, underscoring the urgent need for robust defense strategies.

## 1 Introduction

Imagine a self-driving car navigating the bustling streets of a modern city. It relies on advanced computer vision systems to "see" the world around it—identifying pedestrians, vehicles, road signs, and drivable paths in real time. These systems, powered by deep semantic segmentation models like PSPNet Zhao et al. (2017) and the SegFormer family Xie et al. (2021), have reached impressive accuracy on benchmarks such as Cityscapes Cordts et al. (2016). Yet, beneath this reliability lies a hidden vulnerability: What if a subtle, nearly imperceptible change to the input image could deceive the car into missing something that is actually present? This is the realm of adversarial examples—tiny perturbations crafted to mislead AI models (Szegedy et al., 2014; Goodfellow et al., 2015). Over time, researchers have developed more sophisticated attacks, including iterative optimization methods (Moosavi-Dezfooli et al., 2016; Carlini & Wagner, 2017), and even demonstrated their effectiveness in the physical world, such as tampering with road signs (Metzen et al., 2017; Eykholt et al., 2018).

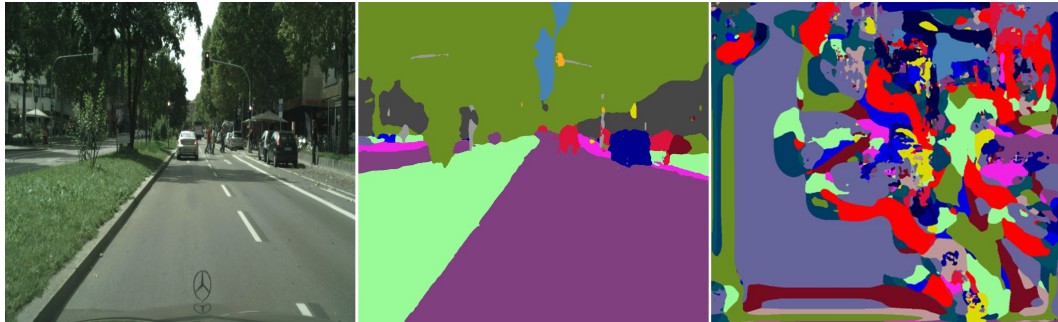

Figure 1: From left to right: the original image, the result under a stealthy attack, and the result under a non-stealthy attack.

The root cause lies in the fact that these deep learning models often rely on fragile and non-robust features that humans cannot perceive, thereby making the models susceptible to adversarial attacks (Ilyas et al., 2019). In autonomous driving, such deceptions could spell disaster, leading to failed obstacle avoidance or misguided path planning.

Building on this, prior research in semantic segmentation attacks has largely focused on global perturbations that aim to disrupt the entire image output, whether in white-box or black-box settings. The key distinction between stealthy and non-stealthy attacks lies in their ability to maintain stable overall predictions while achieving targeted outcomes. As shown in Figure 1, stealthy attacks (the middle figure) subtly manipulate only targeted regions (the cars in the central) or labels in semantic segmentation (Zhong et al., 2022), preserving non-targeted areas to ensure undetectability and realism in scenarios like autonomous driving, such as seamlessly blending vehicles into backgrounds. In contrast, non-stealthy attacks apply broad perturbations that disrupt entire image predictions, for example the right-hand side figure in Figure 1, leading to widespread misclassifications and obvious distortions (Arnab et al., 2018). Recent advancements have refined white-box attacks by incorporating uncertainty-weighted losses (Maag & Fischer, 2024) or approaches like CosPGD for pixel-wise prediction tasks (Agnihotri et al., 2024). However, these global attacks often overlook practical limitations in driving scenarios: an attacker might only target a specific area, like the central road ahead, to maintain stealth and avoid detection. For example, completely scrambling the whole scene could set an alert, but a subtle, localized tweak might quietly mislead the vehicle's detection of drivable areas. Moreover, attackers often lack access to the model's internals, making black-box attacks particularly relevant and dangerous. While defenses like randomized smoothing (Cohen et al., 2019) or input transformations (Xie et al., 2018) offer some protection, they fall short against these constrained, covert threats, and recent uncertainty-based detection approaches highlight the need for more targeted defenses (Xu et al., 2021; Halmosi et al., 2024). Our attack framework can enhance model robustness through adversarial training, as it does not prioritize real-time deployment. Notably, single-modal attacks, can compromise the robustness of multi-modal vision-language-action (VLA) models, causing significant accuracy drops in autonomous systems (Wang et al., 2023). This gap underscores the urgency for understanding regional vulnerabilities and developing robust countermeasures.

In this paper, we propose a region-constrained adversarial attack for autonomous driving, targeting the central image region with inpainting-based pseudo-ground-truth (pseudo-GT) generation. We mask vehicles, inpaint the scene to simulate their absence, and guide the attack with pseudo-GT. The attack uses a two-stage optimization: Stage I employs a composite loss with mean-squared error, total variation, and per-class IoU to align with inpainted outputs and ensure subtle perturbations. Stage II refines with cross-entropy losses to misclassify target pixels while preserving background. Experiments on Cityscapes, KITTI, and GTAV datasets achieve vehicle removal in the target region with over 93% background accuracy under strict constraints. Our contributions include:

- A novel inpainting-guided pseudo-ground-truth mechanism for generating targeted labels in region-constrained segmentation attacks.
- A hybrid loss formulation that combines mean-squared error, total variation, and IoU components to optimize targeted perturbations.

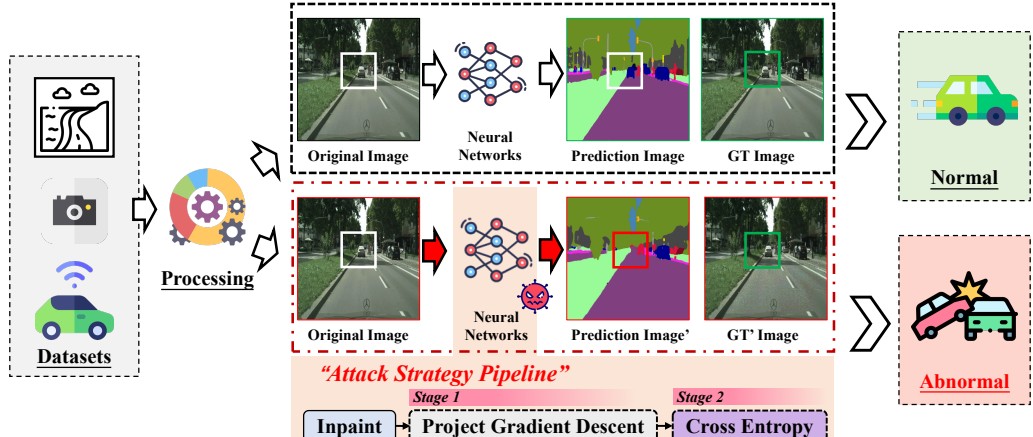

Figure 2: The overview structure of our attack method. This diagram illustrates the two-stage pipeline: pseudo-GT generation via inpainting and perturbation optimization using hybrid losses. It highlights the key components, including mask extraction, inpainting, *Stage I* employs PGD with a composite loss for perturbation synthesis, while *Stage II* uses region-specific cross-entropy for fine-tuning, enabling effective region-constrained black-box attacks.

- A two-stage refinement strategy using region-specific cross-entropy to balance attack potency and background preservation.
- Comprehensive evaluation demonstrating the vulnerability of modern segmentation models under realistic spatial constraints, underscoring the necessity for robust regional defenses.

## 2 RELATED WORK

This section reviews key developments in semantic segmentation, adversarial attacks, region-constrained and black-box methods, and defense strategies relevant to autonomous driving, highlighting gaps our work addresses.

### 2.1 SEMANTIC SEGMENTATION

Semantic segmentation assigns a category to each pixel, critical for autonomous driving. Fully Convolutional Networks (FCN) Long et al. (2015) enabled end-to-end pixel-wise predictions. DeepLab Chen et al. (2018) introduced atrous convolutions for multi-scale context, while PSPNet **?** improved performance with global pyramid pooling. Transformer-based SegFormer Xie et al. (2021) models long-range dependencies, excelling on datasets like Cityscapes. Despite advances, these models remain vulnerable to adversarial attacks.

### 2.2 ADVERSARIAL ATTACKS ON SEMANTIC SEGMENTATION

Adversarial attacks disrupt segmentation with pixel-wise perturbations. Early works (Xie et al., 2017) showed segmentation models like DeepLab and FCN are susceptible to noise (Arnab et al., 2018). Structured attacks optimized segmentation-specific metrics (Cissé et al., 2017). SegPGD Gu et al. (2022) and proximal splitting attacks Rony et al. (2023) improved attack efficacy, while Mao et al. (2020) exposed vulnerabilities in diffusion-based models. These global attacks often ignore practical constraints in driving scenarios.

### 2.3 REGIONAL AND PATCH-BASED ATTACKS

Region-constrained attacks target specific image areas for stealth. Zhong et al. (2022) developed localized attacks preserving non-targeted regions, critical for autonomous driving where central views

are key. Chen et al. (2022) proposed semantically stealthy patches, exploiting object semantics to mislead segmentation while maintaining realism.

## 2.4 BLACK-BOX ATTACKS

Black-box attacks, without model access, use query-based or transfer-based methods. Zeroth-Order Optimization (ZOO) Chen et al. (2020) estimates gradients via finite differences, achieving high success rates. Recent transfer-based approaches Agnihotri et al. (2024) enhance black-box efficacy for segmentation, relevant to real-world threats.

## 2.5 DEFENSE MECHANISMS

Defenses like randomized smoothing (Cohen et al., 2019) offer robustness against $\ell_2$-norm perturbations. Input transformations, such as JPEG compression (Xie et al., 2018), remove noise. Liu et al. (2024) improved robustness using gradient norm regularization. However, defenses often fail against region-constrained or black-box attacks, underscoring the need for targeted solutions.

## 3 METHOD

In this section, we describe our two-stage, region-constrained black-box adversarial attack in detail. We first localize the central vehicle region to construct a binary mask, then generate a pseudo-ground-truth via inpainting, and finally synthesize and refine an $\ell_\infty$-bounded perturbation using a suite of loss functions. The entire pipeline is model-agnostic and treats the target segmentation network as a black box, requiring only forward queries. We summarize the notation in Table 1.

| Symbol | Definition |
|---|---|
| $I \in \mathbb{R}^{H \times W \times 3}$ | input RGB image |
| $M \in \{0,1\}^{H \times W}$ | binary mask of central vehicle pixels |
| $\mathcal{R}$ | rectangular central third region |
| $f(\cdot)$ | black-box segmentation model |
| $\hat{Y} \in \{1, \ldots, C\}^{H \times W}$ | pseudo-ground-truth labels |
| $\delta \in \mathbb{R}^{H \times W \times 3}$ | adversarial perturbation |
| $\epsilon$ | $\ell_\infty$ budget (set to $8/255$) |
| $N_1, N_2$ | iteration counts for Stage I and II |

Table 1: Key notation used in Method.

## 3.1 OVERVIEW

Our method begins by extracting a binary mask $M$ for vehicles in the central third region $\mathcal{R}$ of the input image $I$. We then create an inpainted image $I_{\text{inp}}$ by removing the masked vehicles and querying the model to obtain pseudo-ground-truth labels $\hat{Y}$. In Stage I, we optimize a perturbation $\delta$ using a composite loss to align the adversarial output with $\hat{Y}$. In Stage II, we fine-tune $\delta$ with region-specific cross-entropy losses to enhance misclassification in the target region while preserving the background. The process requires approximately 420 forward queries per image.

## 3.2 CENTRAL REGION MASK AND PSEUDO-GROUND-TRUTH GENERATION

We target only vehicles within the central third of the image. Let

$$\mathcal{R} = \left[ \frac{H}{3}, \ \frac{2H}{3} \right] \times \left[ \frac{W}{3}, \ \frac{2W}{3} \right] \tag{1}$$

be the rectangular crop defining the central region. We obtain an instance segmentation of vehicles (using either a pre-trained detector or ground-truth if available) and define

$$M(p) = \begin{cases} 1, & p \in \mathcal{R} \text{ and pixel } p \text{ is vehicle,} \\ 0, & \text{otherwise.} \end{cases} \tag{2}$$

---

**Algorithm 1** Two-Stage Region-Constrained Black-Box Attack

---

**Require:** Input $I$, mask $M$, black-box model $f(\cdot)$, budget $\epsilon$, steps $N_1, N_2$
**Ensure:** Adversarial example $I^{\text{adv}}$
1: $I_{\text{inp}} \leftarrow \text{Inpaint}(I, M)$
2: $\hat{Y} \leftarrow \arg\max f(I_{\text{inp}})$                                         {pseudo-GT}
3: Initialize $\delta \sim \mathcal{U}[-\epsilon, \epsilon]$
4: **for** $t = 1$ **to** $N_1$ **do**
5:     Compute $\mathcal{L}_1$ via Eq. 5
6:     $\delta \leftarrow \text{AdamStep}(\nabla\mathcal{L}_1)$
7:     Clip $\delta \leftarrow \text{Proj}_{[-\epsilon,\epsilon]}(\delta)$
8: **end for**
9: $I^{(1)} \leftarrow I + \delta$
10: **for** $t = 1$ **to** $N_2$ **do**
11:     Compute $\mathcal{L}_2$ via Eq. 8
12:     $\delta \leftarrow \text{AdamStep}(\nabla\mathcal{L}_2)$
13:     Clip $\delta \leftarrow \text{Proj}_{[-\epsilon,\epsilon]}(\delta)$
14: **end for**
15: **return** $I^{\text{adv}} = I + \delta$

---

This binary mask $M$ localizes the attack region, enforcing spatial constraints on the perturbation.

To guide the attack toward misclassifications, we simulate the model's output when the vehicle is absent. Denote by $\text{Inpaint}(\cdot)$ a mask-aware inpainting operator (e.g., Telea's algorithm with radius 5). We compute

$$I_{\text{inp}} = \text{Inpaint}(I, M), \tag{3}$$

replacing vehicle pixels in $\mathcal{R}$ with contextually plausible background. We then query the black-box model:

$$\hat{P} = f(I_{\text{inp}}) \in \mathbb{R}^{C \times H \times W}, \quad \hat{Y}(p) = \arg\max_c \hat{P}_c(p). \tag{4}$$

By construction, $\hat{Y}$ assigns non-vehicle labels within $\mathcal{R}$ (e.g., road, sidewalk) while matching the model's prediction on $I$ outside $\mathcal{R}$. This pseudo-ground-truth $\hat{Y}$ serves as our target for adversarial optimization.

## 3.3 STAGE I: PERTURBATION SYNTHESIS

We aim to find $\delta$ with $|\delta|_\infty \leq \epsilon$ ($\epsilon = 8/255$), where the larger perturbation budget is chosen to enhance attack efficacy while balancing perceptibility constraints. We initialize $\delta^{(0)} \sim \mathcal{U}[-\epsilon, \epsilon]$ and perform $N_1 = 300$ iterations of Adam updates on the following composite loss:

$$\mathcal{L}_1 = \underbrace{\frac{1}{HW} \sum_p \left\| f(I+\delta)_p - \mathbf{e}_{\hat{Y}(p)} \right\|_2^2}_{\mathcal{L}_{\text{MSE}}} + \lambda_{\text{IoU}} \underbrace{\sum_{c \neq c_{\text{veh}}} \left(1 - \text{IoU}_c\right)}_{\mathcal{L}_{\text{IoU}}} + \lambda_{\text{TV}} \underbrace{\sum_p |\nabla(I+\delta)(p)|}_{\mathcal{L}_{\text{TV}}}, \tag{5}$$

where $\mathbf{e}_k \in \mathbb{R}^C$ is the one-hot vector for class $k$, $\lambda_{\text{IoU}} = 5.0$, and $\lambda_{\text{TV}} = 0.5$. The IoU for class $c$ is

$$\text{IoU}_c = \frac{\sum_p P_c(p) \mathbf{1}(\hat{Y}(p)=c)}{\sum_p \left[P_c(p) + \mathbf{1}(\hat{Y}(p)=c)\right] - \sum_p P_c(p) \mathbf{1}(\hat{Y}(p)=c)}, \tag{6}$$

with $P_c(p) = \text{softmax}\big(f(I+\delta)_p\big)_c$. The total variation (TV) is computed on the adversarial input $I + \delta$ as

$$\nabla(I+\delta)(p) = \left|(I+\delta)_p - (I+\delta)_{p+\hat{x}}\right| + \left|(I+\delta)_p - (I+\delta)_{p+\hat{y}}\right|. \tag{7}$$

Each iteration updates $\delta$ with step size $\alpha = 32/255$ and projects to $[-\epsilon, \epsilon]$. After $N_1$ iterations, we obtain $\delta^{(1)*}$ and $I^{(1)} = I + \delta^{(1)*}$.

## 3.4 STAGE II: REGION-SPECIFIC CE FINE-TUNING

Stage I aligns predictions with $\hat{Y}$, but may alter background slightly. We perform $N_2 = 120$ steps to enforce region-specific behavior. Let $O = f(I^{(1)}) \in \mathbb{R}^{C \times H \times W}$. We minimize:

$$\mathcal{L}_2 = \underbrace{\sum_{p:\, M(p)=1} \mathrm{CE}(O_p, \hat{Y}(p))}_{\mathcal{L}_{\mathrm{CE}}^{veh}} + \lambda_{bg} \underbrace{\sum_{p:\, M(p)=0} \mathrm{CE}(O_p, f(I)_p)}_{\mathcal{L}_{\mathrm{CE}}^{bg}}, \tag{8}$$

where $\mathrm{CE}(O_p, k) = -\log\left(\frac{\exp(O_p[k])}{\sum_{c=1}^{C} \exp(O_p[c])}\right)$, and $\lambda_{bg} = 0.5$. We optimize $\delta$ (initialized from $\delta^{(1)*}$) with Adam and clip to $\|\delta\|_\infty \leq \epsilon$. The final $I^{\mathrm{adv}} = I + \delta^{(2)*}$.

## 3.5 ALGORITHM AND COMPLEXITY

Algorithm 1 summarizes the pipeline. Each image requires $N_1 + N_2 = 420$ forward-backward calls to the black-box model on a single NVIDIA 4070ti super GPU, the end-to-end process (including inpainting) runs at 1024×1024 resolution.

# 4 EXPERIMENTS

## 4.1 DATASETS

The primary dataset is Cityscapes (Cordts et al., 2016), containing 5,000 annotated urban scene images from 50 cities with 19 semantic classes, focusing on safe driving impacts. For generalization, we evaluate on the synthetic GTA5 dataset (Richter et al., 2016), with 24,966 images under diverse conditions (e.g., day, night, dusk) and compatible 19 classes. Additionally, the KITTI Semantic Segmentation dataset (Alhaija et al., 2018) is used, consisting of 200 training images with pixel-level annotations for 19 classes in real-world driving scenes from a vehicle-mounted camera.

## 4.2 METRICS

We evaluate the attack's efficacy and stealthiness using four metrics. *MaskAcc* measures the accuracy of adversarial segmentation in the targeted region against the pseudo-ground-truth. *BgAcc* assesses the preservation of background pixels relative to original predictions. *PSNR* and *SSIM* quantify perturbation perceptibility by comparing the original and adversarial images. Detailed formulations are provided in Appendix A.2.

## 4.3 IMPLEMENTATION DETAILS

The method is implemented using the PyTorch framework, requiring minimal GPU resources and tested on an NVIDIA RTX 4070 Ti. Inpainting employs OpenCV's `inpaint` (Telea method, radius 5). The semantic segmentation models are SegFormer (Xie et al., 2021), DeepLabV3+ with ResNet101 backbone (Chen et al., 2018), and HRNetV2-W48 (Yuan et al., 2020). Input images are resized to 1024×1024. Evaluations use the Cityscapes validation set, targeting class 13 ("car"). Perturbations are confined to the central horizontal one-third for driving safety focus, with 300 PGD steps, 120 CE refinement steps, and maximum perturbation magnitude $\epsilon$=8.

## 4.4 QUANTITATIVE ANALYSIS

The following results evaluate our attack method on three segmentation models: HRNetV2-W48, DeepLabV3+, and SegFormer. As shown in Table 2, our proposed method achieves high pixel-level accuracy. For instance, HRNetV2-W48 attains 95.40% MaskAcc, indicating precise mask generation for adversarial images, and 88.58% BgAcc, reflecting effective preservation of background pixels.

Figure 3: The three key steps of our proposed method. The first column displays the original image alongside its semantic segmentation results. The second column presents the inpainted image and its corresponding segmentation output, generated by masking and inpainting the central vehicle region. The third column shows the perturbation overlayed adversarial image and its segmentation results, illustrating the effect of the optimized perturbation.

| Model | MaskAcc (%) | BgAcc (%) |
|---|---|---|
| HRNetV2-W48 | 95.40 | 88.58 |
| DeepLabV3+ (ResNet101) | 91.82 | 92.01 |
| SegFormer | 92.78 | 93.42 |

Table 2: Performance of our method on the Cityscapes dataset for vehicle removal attacks

Results across models and scenarios show that, for DeepLabV3+ (ResNet101) and SegFormer, generating a random square area with a random class is simpler than inpainting and creating fake masks. However, MaskAcc for HRNetV2-W48 drops by about 3%. The attack uses a single random initialization within the $\epsilon$ budget, leading to variability across iterations for the same image. Compared to DeepLabV3+ and SegFormer, HRNetV2-W48 maintains a dedicated high-resolution branch, making it more reliant on fine-grained local cues. Thus, pixel-level perturbations impact HRNet more, resulting in lower background accuracy under identical attack settings.

| Model | MaskAcc (%) | BgAcc (%) |
|---|---|---|
| HRNetV2-W48 | 90.04 | 89.07 |
| DeepLabV3+ (ResNet101) | 91.49 | 91.50 |
| SegFormer | 92.62 | 92.87 |

Table 3: Performance of our method on different models with the GTAV dataset.

We also tested on GTAV to assess generality. As shown in Table 3, our method yields high MaskAcc and BgAcc on GTAV, demonstrating precision across varied scenarios and visual styles.

| Model | MaskAcc (%) | BgAcc (%) |
|---|---|---|
| HRNetV2-W48 | 94.57 | 98.64 |
| DeepLabV3+ (ResNet101) | 94.97 | 96.70 |
| SegFormer | 92.83 | 94.49 |

Table 4: Average results on the KITTI dataset across different models

To further validate the generalization of our method, we evaluated it on the KITTI Semantic Segmentation dataset. Table 4 summarizes the average MaskAcc and BgAcc across the tested models

(HRNet, DeepLabV3+, and SegFormer), excluding cases where MaskAcc=0 due to limited vehicle pixels. The results show high performance, with MaskAcc exceeding 92% and BgAcc above 94% on average, demonstrating the method's effectiveness on real-world driving scenes. The better performance on KITTI compared to Cityscapes may be attributed to KITTI's smaller scale and more consistent scenes (e.g., uniform lighting and fewer environmental variations from German roads), which allow for more stable perturbation effects under our central region constraint.

A comprehensive evaluation across multiple scenarios and datasets (Cityscapes, GTAV, and KITTI) indicates that our method consistently achieves high accuracy in refining the final semantic segmentation while posing a significant threat to visually natural images. Compared to baselines like SegPGD Gu et al. (2022), our region-constrained attack maintains higher background preservation (BgAcc > 93% in most cases) while effectively degrading target segmentation, underscoring its stealthiness and broad applicability.

## 4.5 PERTURBATION PERCEPTIBILITY

| Metric | Average Value (± Std. Dev.) |
|---|---|
| PSNR (dB) | 51.13 (±0.01) |
| SSIM | 0.9979 (±0.0003) |

Table 5: Average PSNR and SSIM for adversarial perturbations on GTAV images. High values indicate low perceptibility.

To assess perturbation perceptibility, we compute PSNR and SSIM between original and adversarial images on the GTAV dataset. Table 5 shows an average PSNR of 51.13 dB (±0.01 dB) and SSIM of 0.9979 (±0.0003), indicating nearly imperceptible perturbations, with PSNR well above the 30 dB threshold and SSIM near 1, ensuring structural preservation (Rony et al., 2023). This stealthiness is critical for autonomous driving, enabling effective targeted degradation without visual detection.

## 4.6 ABLATION STUDY

The PGD and CE steps can significantly influence overall performance but may increase computation time. As shown in Table 6 for HRNetV2-W48, PGD steps have a lesser impact during training, as they primarily modify the perturbation. In contrast, CE plays a more pronounced role: without CE steps, MaskAcc declines by about 25% (from 92.60% to 65.19%).

| Primary–Secondary Steps | MaskAcc (%) | BgAcc (%) |
|---|---|---|
| 50–200 | 89.12 | 83.02 |
| 100–200 | 90.83 | 82.87 |
| 200–200 | 91.96 | 82.71 |
| 300–0 | 65.19 | 89.74 |
| 300–50 | 91.10 | 80.05 |
| 300–100 | 91.51 | 84.87 |
| 300–200 | 92.60 | 88.51 |

Table 6: Ablation study of HRNetV2-W48 under different primary–secondary parameter combinations. MaskAcc and BgAcc denote mask-pixel and background-pixel accuracy, respectively.

Increasing CE steps to 50 improves MaskAcc but reduces BgAcc by around 9%, indicating that CE harmonizes performance between the attack and background areas. Further increasing CE from 100 to 200 yields minimal gains, suggesting that performance stabilizes.

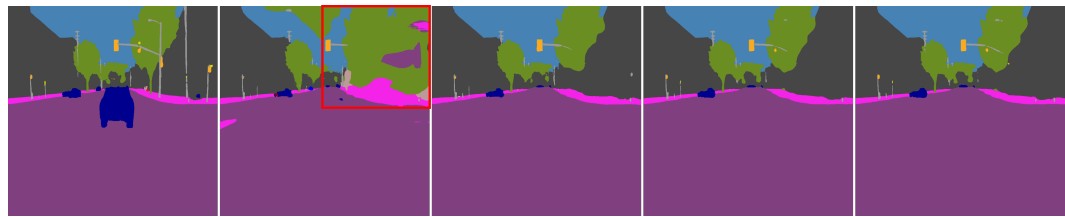

Figure 4: Visualization of our method's performance under different maximum perturbation levels. The first image shows the ground truth. The remaining images show maximum perturbations at 1, 4, 8, and 16.

| Max Perturbation Level | MaskAcc (%) | BgAcc (%) |
|---|---|---|
| 0.1 | 36.32 | 73.95 |
| 0.5 | 37.87 | 75.95 |
| 1 | 51.54 | 77.11 |
| 4 | 83.37 | 85.32 |
| 8 | 91.54 | 91.17 |
| 16 | 91.89 | 92.98 |

Table 7: Ablation study of the attack method with SegFormer on the Cityscapes dataset under different maximum perturbation levels.

The perturbation level also affects performance, influencing both semantic predictions and visual perceptibility. Results from the perturbation ablation study (Table 7) on Cityscapes with SegFormer show that levels below 1 severely compromise final prediction accuracy, while a level of 4 yields optimal results in most images. Levels above 8 have negligible further impact.

| Max Perturbation Level | MaskAcc (%) | BgAcc (%) |
|---|---|---|
| 0.1 | 34.13 | 82.44 |
| 0.5 | 40.43 | 83.59 |
| 1 | 56.68 | 84.65 |
| 4 | 88.41 | 93.94 |
| 8 | 92.04 | 96.49 |
| 16 | 94.37 | 97.96 |

Table 8: Attack method with SegFormer on the GTAV dataset under different perturbation levels.

On GTAV (Table 8), performance is more stable, as shown in Figure 4, with negligible differences between levels 4, 8, and 16, and level 1 still yielding satisfactory results, although the overall accuracy is a little bit low, the overall visualization of the predictions is still acceptable. We observe that GTAV is less challenging than Cityscapes due to its synthetic nature, featuring simpler shadows, consistent lighting, and minimal noise, enabling models like SegFormer, HRNet, and DeepLab to achieve stable performance with lower perturbation levels (e.g., level 4 suffices for good results).

## 5 CONCLUSION

We proposed a pseudo-GT driven region-constrained black-box attack framework for semantic segmentation in autonomous driving. By integrating inpainting for pseudo-ground-truth generation and two-stage optimization with hybrid losses, our method hides targets in the background with high stealthiness. Evaluations on Cityscapes, GTAV, and KITTI datasets show superior performance and nearly imperceptible perturbations. Ablation studies confirm key components' roles, highlighting model vulnerabilities and the need for robust defenses. Future work could extend to physical-world or multi-modal attacks.

## 6 ETHICS STATEMENT

This work adheres to the ICLR Code of Ethics. Our research focuses on adversarial attacks in semantic segmentation models, which could potentially be misused for malicious purposes, such as disrupting autonomous driving systems or generating misleading visual interpretations. To mitigate harm, we emphasize that the proposed method is intended for defensive research and improving model robustness. No human subjects were involved, and the experiments utilize publicly available datasets (e.g., Cityscapes) with no privacy concerns. The work is self-funded without external sponsorship.

## 7 REPRODUCIBILITY STATEMENT

To ensure reproducibility, we provide comprehensive details in the main paper and supplementary materials. The core algorithm, including inpainting methods (TELEA, NS, and biharmonic), PGD attack with soft alignment, and cross-entropy refinement, is described in Sections 3 and 4, with hyperparameters listed in Table 1. Datasets (Cityscapes) are publicly accessible.

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

# A APPENDIX

## A.1 THE USE OF LARGE LANGUAGE MODELS (LLMs)

In this work, we utilized large language models (LLMs), as a general-purpose assistive tool for grammar checking, sentence rephrasing, and minor editing to improve clarity and flow in the manuscript. The LLM was not involved in generating core research ideas, experimental designs, results analysis, or conclusions; all scientific content was authored by the human researchers. We manually reviewed and edited all LLM-suggested changes to ensure accuracy, originality, and alignment with our intended meaning. The authors take full responsibility for the entire content of this paper, and no LLMs are considered authors or co-contributors.

## A.2 METRICS FORMULATIONS

MaskAcc measures the accuracy of the adversarial segmentation mask in the inpainted attack region compared to the pseudo-ground-truth:

$$\text{MaskAcc} = \frac{\sum_{p \in \mathcal{M}} \mathbf{1}(\hat{y}_p = y_p^{\text{pseudo}})}{|\mathcal{M}|} \times 100\%, \tag{9}$$

where $\mathcal{M}$ is the set of pixels in the attacked region, $\hat{y}_p$ is the adversarial prediction at pixel $p$, and $y_p^{\text{pseudo}}$ is the pseudo-ground-truth label.

BgAcc assesses the preservation of background pixels (those unaffected by the attack) relative to the original predictions:

$$\text{BgAcc} = \frac{\sum_{p \notin \mathcal{M}} \mathbf{1}(\hat{y}_p = y_p^{\text{orig}})}{H \times W - |\mathcal{M}|} \times 100\%, \tag{10}$$

where $y_p^{\text{orig}}$ is the original prediction at pixel $p$, and $H \times W$ is the image size.

To evaluate perturbation perceptibility, we compute Peak Signal-to-Noise Ratio (PSNR) between the original image $I$ and adversarial image $I^{\text{adv}}$:

$$\text{PSNR} = 10 \log_{10} \left( \frac{\text{MAX}^2}{\text{MSE}(I, I^{\text{adv}})} \right), \tag{11}$$

where MAX = 255 for 8-bit images, and MSE is the mean squared error.

Structural Similarity Index Measure (SSIM) is defined as:

$$\text{SSIM}(I, I^{\text{adv}}) = \frac{(2\mu_I \mu_{I^{\text{adv}}} + c_1)(2\sigma_{II^{\text{adv}}} + c_2)}{(\mu_I^2 + \mu_{I^{\text{adv}}}^2 + c_1)(\sigma_I^2 + \sigma_{I^{\text{adv}}}^2 + c_2)}, \tag{12}$$

where $\mu$ and $\sigma$ are mean and variance, $\sigma_{II^{\text{adv}}}$ is covariance, and $c_1, c_2$ are constants for stability.

## A.3 PERFORMANCE WITH DIFFERENT INPAINT METHOD

| Max Perturbation Level | MaskAcc (%) | BgAcc (%) |
|---|---|---|
| CV2.TELEA | 93.97 | 95.70 |
| CV2.NS | 92.13 | 94.11 |
| Biharmonic | 93.93 | 95.53 |

Table 9: Performance of the DR model on the GTAV dataset using different inpaint methods.

The different inpaint methods (NS, Biharmonic, and TELEA) exhibit similar performance in terms of MaskAcc and BgAcc on the GTAV dataset, with minimal differences, indicating that these methods have limited impact on pixel-level accuracy.

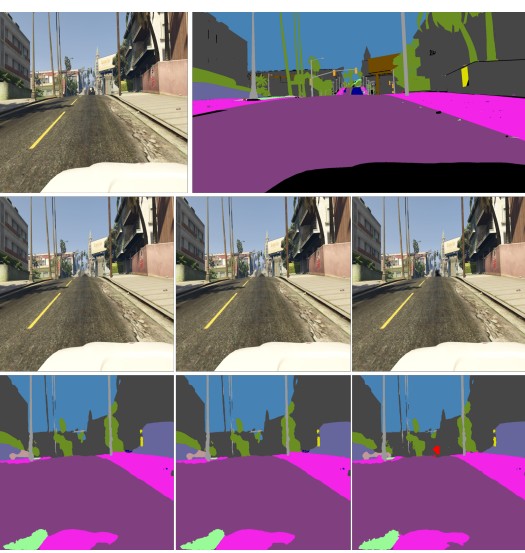

Figure 5: The first row shows the original image (left) and its ground truth semantic segmentation map (right). The second row displays the inpainted images from left to right using TELEA, NS, and biharmonic methods, respectively. The third row presents the corresponding attack results on the semantic segmentation maps induced by these inpainting methods.

Different inpainting methods do not affect the accuracy rates of the attack approach, but they do influence the final visual effects in the segmentation maps. For instance, the biharmonic method may cause the repaired regions to be interpreted by the model as red pedestrian areas.

