# OpenReview forum: "Pseudo-GT Driven Region-Constrained Black-Box Attack on Semantic Models for Autonomous Driving"
_ICLR.cc/2026/Conference — ICLR 2026 Conference Withdrawn Submission_

### Official Review · Reviewer_f5hn · 2025-10-15

**Soundness:** 1
**Presentation:** 2
**Contribution:** 1
**Rating:** 2
**Confidence:** 5

**Summary:**

This paper constrains the adversarial perturbation to the central region of the input image. Using inpainting, the pixels of the vehicle within the central region are erased and filled with pixels of the surrounding environment (such as the road and sky), whose semantic segmentation result is then used to supervise the optimization of the central region perturbation.

**Strengths:**

1. This paper enhances the attack stealthiness by constraining the adversarial perturbation to the central region and maintaining the prediction results of other regions.

2. The authors evaluate across three datasets (Cityscapes, KITTI, GTAV) and three different segmentation models (SegFormer, DeepLabV3+, HRNet), and they report multiple metrics

**Weaknesses:**

1. This article considers the autonomous driving (AD) scenario, but the proposed attack is still digital and cannot be deployed in the real world. Therefore, it cannot threaten the actual application of AD.

2. The optimized perturbation is input-specific and not universal. This means that perturbations need to be generated in real time as the vehicle moves. However, this paper did not discuss the run time. In fact, the method requires ~420 queries per image. Such a query budget would be impractical for attacking a running AD system.

3. The proposed method is largely a repackaging of well-established components—region masking, inpainting, PGD-style iterative optimization, and standard IoU/TV regularizers. While these are assembled into a pipeline, the conceptual advance beyond prior localized or stealthy adversarial attacks is minimal. The contribution feels more like an incremental engineering variation rather than a substantive new idea.

4. There are currently a large number of global attacks and local attacks, but this paper does not consider baselines for comparison in the experiment.

5. Figure 2 is confusing: Its design initially misled me into thinking that the paper was addressing a poisoning or backdoor attack, rather than an adversarial attack.

6. There is no visualization of the crafted perturbations in experiments.

**Questions:**

None

---

### Official Review · Reviewer_U52G · 2025-10-27

**Soundness:** 2
**Presentation:** 1
**Contribution:** 1
**Rating:** 0
**Confidence:** 4

**Summary:**

The paper introduces a two-stage adversarial attack on semantic segmentation for autonomous driving. The method crafts localized perturbations on the central vehicle region using hybrid and region-specific losses to mislead predictions while preserving the background. Experiments on Cityscapes with SegFormer show high attack success, revealing that spatial constraints do not guarantee robustness.

**Strengths:**

N/A

**Weaknesses:**

The paper is not well-written and lacks sufficient comparisons with existing works in the field.
For instance, examples of relevant prior works include:
[A] Pietrosanti et al., “Benchmarking the Spatial Robustness of DNNs via Natural and Adversarial Localized Corruptions,” Pattern Recognition, 2025.
[B] Rossolini et al., “On the Real-World Adversarial Robustness of Real-Time Semantic Segmentation Models for Autonomous Driving,” IEEE TNNLS, 2023.
[C] K. K. Nakka and M. Salzmann, “Indirect Local Attacks for Context-Aware Semantic Segmentation Networks,” ECCV, 2020.
[D] A. Arnab et al., “On the Robustness of Semantic Segmentation Models to Adversarial Attacks,” CVPR, 2018.

Other weaknesses:
- The introduction is written from a very general perspective, starting by reintroducing the concept of adversarial examples from scratch rather than focusing on the specific technical problem. It lacks depth and technical clarity.
- In Figure 1, the paper should include a clear visual representation of the input perturbation. This would help the reader understand, from the beginning, the distinction between stealthy and non-stealthy attacks.
- The related work section completely omits key references, such as those mentioned above, and these works are not discussed anywhere in the main text.
- The paper is poorly structured, lacks proper experimental comparisons, and provides an insufficient discussion of defense mechanisms. Moreover, the authors do not seem to have evaluated these defenses in depth.

**Questions:**

- Did the authors test any defense mechanisms against the proposed attacks? If not, why were they mentioned in the related work section?

- How is this work relevant compared to previous studies on adversarial attacks for semantic segmentation in driving?

---

### Official Review · Reviewer_BgSG · 2025-10-28

**Soundness:** 2
**Presentation:** 2
**Contribution:** 2
**Rating:** 2
**Confidence:** 4

**Summary:**

Semantic segmentation models are critical for applications like autonomous driving but remain highly susceptible to adversarial perturbations. The authors introduce a two-stage region-based attack that targets the central vehicle area while preserving background semantics. First, a pseudo-ground-truth mask is created by inpainting the vehicle region to simulate its absence. Then, an ℓ∞-bounded perturbation is optimized using a hybrid loss combining MSE, total variation, and class-wise IoU terms, followed by region-specific cross-entropy refinement. On Cityscapes, the attack achieves over 92% accuracy within the target zone and 93% background preservation on SegFormer, demonstrating that spatial constraints offer limited defense against structured, localized attacks.

**Strengths:**

- The authors address an important and security-related topic.
- It's good that three different datasets were tested.
- The additional metrics PSNR and SSIM used are good.
- The models used are well-known and frequently applied; it is beneficial that both CNNs and transformers were utilized.
- I find it beneficial that ablation studies are being conducted.

**Weaknesses:**

- "recent uncertainty-based detection approaches highlight the need for more targeted defenses (Xu et al., 2021; Halmosi et al., 2024)." Xu introduces divide-and-conquer adversarial training and Halmosi re-evaluate a number of well-known robust segmentation models in an extensive empirical analysis - so no uncertainty. Uncertainty-based detection occurs here: "Uncertainty-based Detection of Adversarial Attacks in Semantic Segmentation" Maag et al.
- In Figure 2, GT image is more of an RGB image than a mask.
- In related work, the section on semantic segmentation is very brief/incomplete and not necessarily relevant here, as it deals with attacks and not with developing new models.
- The related work section is incomplete. E.g.
1. Section 2.3.: “Universal Adversarial Perturbations Against Semantic Image Segmentation” Metzen et al., as well as other targeted attacks such as FGSM, PGD, and DAG, can be trained so that, for example, only car pixels are converted into roads. "Chen et al. (2022) proposed semantically stealthy patches" They do not use patches.
2. In Section 2.4, "Recent transfer-based approaches Agnihotri et al. (2024) enhance black-box efficacy" Agnihotri present white-box adversarial attacks.
3. In section 2.5, the defense/detection methods should also be for semantic segmentation, e.g. "Improved Noise and Attack Robustness for Semantic Segmentation by Using Multi-Task Training with Self-Supervised Depth Estimation" Klingner et al,
"Characterizing Adversarial Examples Based on Spatial Consistency Information for Semantic Segmentation" Xiao et al,
"Certified Defences Against Adversarial Patch Attacks on Semantic Segmentation" Yatsura et al.
- The paper is not written very clearly. For example, there is no source reference for Telea's algorithm in section 3.2, the heading in section 4 is blue, and references to the appendix are incomplete.
- Accuracy as the sole metric is less meaningful than mIoU, as large classes simply dominate.
- There were no baseline/state-of-the-art comparisons with other attacks.

**Questions:**

- In section 2.3, is the point here that not all pixels receive noise, as is the case with patch attacks, for example, in “Evaluating the Robustness of Semantic Segmentation for Autonomous Driving
against Real-World Adversarial Patch Attacks” by Nesti et al, or that not all pixels are attacked in the prediction? This is unclear here.
- Why is the number of forward queries per image so high, and doesn't that cause the runtimes to increase dramatically?
- "We obtain an instance segmentation of vehicles" Why use instance segmentation rather than semantic segmentation if no distinction is made between instances in the further course?
- Adamstep is used and the gradient of the loss function is determined, so access to the model is required and therefore a whitebox setting? In addition, the method requires access to the logits, which corresponds more to a gray box than a black box.
- What are the training/test splits for the datasets? Because the three models used were each trained on each dataset, right?
- "Results across models and scenarios show that, for DeepLabV3+ (ResNet101) and SegFormer, generating a random square area with a random class is simpler than inpainting and creating fake masks." Yes, that makes sense. Was this compared or done somewhere?
- Why does background accuracy increase with the perturbation level? I would actually say that the stronger the attack, the more likely it is to affect background pixels.

---

### Official Review · Reviewer_ktsQ · 2025-10-31

**Soundness:** 2
**Presentation:** 2
**Contribution:** 1
**Rating:** 2
**Confidence:** 4

**Summary:**

This work explores the black-box attack on semantic segmentation task, by proposing a two-stage query-based method. In particular, detected vehicle masks are filled into one-third of the input image region to generate a pseudo ground-truth segmentation image. Meanwhile, mean squared error loss, spatial consistency loss, and class intersection-over-union loss are employed to misclassify the target category. Then, region-specific cross-entropy loss is used to mislead vehicle pixels toward surrounding categories while preserving background consistency.

**Strengths:**

1. To achieve stealthy attacks, the authors introduce a combination of mean squared error loss, spatial consistency loss, class intersection-over-union loss, and region-specific cross-entropy loss to jointly optimize the adversarial perturbations. The idea seems nice.

2. This manuscript is generally well-structured and easy to read.

**Weaknesses:**

1. Lack of comparison with other query-based black-box attack methods. The experiments in the paper only compare different variants of the proposed method itself, without including other existing query-based black-box attack approaches. If there are currently no query-based black-box attacks specifically designed for semantic segmentation, the authors could introduce and compare with traditional image-level query-based methods as baselines.
2. Limited evaluation across different scenarios. The authors specifically target the autonomous driving scenario, performing stealthy attacks on vehicles located in the front one-third region of the image. However, they should also consider attacking other critical objects such as pedestrians or other potential safety hazards to demonstrate broader applicability and effectiveness.
3. Unclear motivation. The authors primarily rely on pre-generated pseudo-label masks and apply a targeted adversarial attack to achieve stealthiness. However, their key contribution appears to lie in the generation of these pseudo-labels, while the rationale and justification for combining multiple loss functions in the subsequent stages are not clearly explained or grounded in solid methodology.
4. Insufficient ablation studies and visualization. The authors lack visual comparisons demonstrating the impact of different loss functions through ablation studies. Additionally, important evaluation metrics such as computational time cost (e.g., frames per second, FPS) are missing. Furthermore, the authors should provide results showing how attack effectiveness evolves with the number of queries to better evaluate query efficiency.
5. There are many semantic segmentation benchmarks, but this work only focuses on Cityscapes and a little on KITTI, which is insufficient to provide solid experimental results.
6. The implementation details are missing without any code attached.

**Questions:**

See the weaknesses.

---

### Note · Authors · 2025-11-21

I have read and agree with the venue's withdrawal policy on behalf of myself and my co-authors.